

# Measurement report: On the contribution of long-distance transport to the secondary aerosol formation and aging

Haobin Zhong[1,2], Ru-Jin Huang[1,2,3], Chunshui Lin[1], Wei Xu[1], Jing Duan[1], Yifang Gu[1,2], Wei Huang[1], Haiyan Ni[1], Chongshu Zhu[1], Yan You[4], Yunfei Wu[5], Renjian Zhang[5], Jurgita Ovadnevaite[6], Darius Ceburnis[6], Colin D. O'Dowd[6]

[1]State Key Laboratory of Loess and Quaternary Geology (SKLLQG), Center for Excellence in Quaternary Science and Global Change, and Key Laboratory of Aerosol Chemistry and Physics, Institute of Earth Environment, Chinese Academy of Sciences, Xi'an 710061, China

[2]University of Chinese Academy of Sciences, Beijing 100049, China

[3]Open Studio for Oceanic-Continental Climate and Environment Changes, Pilot National Laboratory for Marine Science and Technology (Qingdao), 266061 Qingdao, China

[4]National Observation and Research Station of Coastal Ecological Environments in Macao, Macao Environmental Research Institute, Macau University of Science and Technology, Macao SAR 999078, China

[5]Key Laboratory of Middle Atmosphere and Global Environment Observation (LAGEO), Institute of Atmospheric Physics, Chinese Academy of Sciences, Beijing 100029, China

[6]School of Physics and Ryan Institute's Centre for Climate & Air Pollution Studies, National University of Ireland Galway, University Road, Galway H91CF50, Ireland

Correspondence to: Ru-Jin Huang (rujin.huang@ieecas.cn)

## Abstract

To investigate the physio-chemical properties of aerosol transported from major pollution regions in China, observations were conducted ~200 m above the ground at the junction location of the North China Plain and Fenwei Basin, which are two regions of top priority for China's blue sky campaign. We identified three pollution transport sectors including those from Beijing-Tianjin-Hebei (BTH), urban Guanzhong Basin (GZB), northern China and one clean transport sector from rural Guanzhong Basin region. Secondary inorganic aerosol (SIA) constituted a major fraction (39-46%) in all pollution transport sectors with high sulphur oxidation ratio (0.44-0.58) and nitrogen oxidation ratio (0.24-0.29), suggesting efficient formation of secondary inorganic aerosol during regional transport. While more oxidized oxygenated organic aerosol (MO-OOA) played a dominant role in all sectors including the clean one, accounting for 42-58% of total organic aerosol. Elemental analysis (O and C) shows that aerosol particles at this receptor site were much more oxidized than urban regions, pointing that long-range transport contributed markedly to the organic aerosol oxidation and aging. Case studies of pollution events with high sulphate, nitrate and more-oxidized oxygenated organic aerosol production rate indicate the strong formation efficiency of secondary aerosol during regional transport in the Beijing-Tianjin-Hebei transport sector.



**Keywords:** Regional transport; Secondary aerosol formation; More oxidized organic aerosol; Air pollution.

**1 Introduction**

Air pollution events with high levels of fine particles (particulate matter with a diameter $\leq 2.5$ µm, $PM_{2.5}$) were frequently occurred in China over the past years, due to rapid industrialization and urbanization (Lelieveld et al., 2015; Feng et al., 2018; An et al., 2019). The high level of $PM_{2.5}$ affects air quality, human health and climate, thus, has received widespread concerns around the world (Tie et al., 2016; Cohen et al., 2017). To better understand air pollution in China, many field studies has been carried out in the last decades (Tie et al., 2009; Lei et al., 2011; Cao et al., 2012; Huang et al., 2014). Most of these studies for particle properties are based on local observations, such as in Beijing (Sun et al., 2013; Li et al., 2019), Shanghai (Xu et al., 2012; Huang et al., 2013; Wang et al., 2020), Xi'an (Huang et al., 2014; Duan et al., 2021; Lin et al., 2022), Guangzhou (Guo et al., 2020; Chen et al., 2021), and Hong Kong (Li et al., 2015; Sun et al., 2016). However, aerosol particles can affect hundreds of kilometers through transport depending on particle size and chemical compositions (Uno et al., 2009). During transport, aerosols undergo further transformation, altering chemical composition and oxidation level and consequently affecting their chemo-physical properties and climate impact (Moffet and Prather, 2009; Riemer and West, 2013; Calvo et al., 2013; Fierce et al., 2016).

Recent studies found that local formation cannot fully explain the increase of SIA during pollution events, and the regional transport was considered as an important source for the increase of SIA (Yang et al., 2015; Tang et al., 2016). Some modeling studies reported that heterogeneous chemistry during the transport was identified as the dominant factor during haze episodes in mega cities (Li and Han, 2016; Li et al., 2017), and were further supported by the observations. Du et al. (2019) reported that the chemical transformation from $SO_2$ to sulphate was the major source of sulphate in Beijing. Li et al. (2021) suggested that the pollution in winter in Beijing was largely affected by the regional transport, and the water vapor during the transport of the air mass greatly increased SIA proportion. Gunsch et al. (2018) claimed that the particles were heavily coated with SOA formed during the transport, with 89% of organics fractions in $PM_1$ and 0.8 O/C ratio in the forested Great Lakes region during wild-fire period. Most of the existing studies were devoted to studying the contribution of regional transport to pollution events in urban areas, while the study on region-to-region transport was limited. Our previous study reported that different regions in China represented different chemical compositions and OA sources due to different types of emission characteristics (Zhong et al., 2020). Therefore, the transport aerosol particles from different regions may have completely different properties due to different precursors and transport conditions. The study of region-to-region transport can provide insight to the interactions and mixing properties of particles on a national scale.

Investigation of the chemical compositions and sources with the transport pathways in background areas is a common method to understand the influence of long-distance transport of aerosol on the atmospheric environment (Schichtel et al., 2006; Salvador et al., 2008; Das and Jayaraman, 2012; Tang et al., 2014; Pu et al., 2015). In this study, we performed a two-



months observation at a regional receptor site to investigate the characteristics of aerosol
transported from the major pollution regions by using a time-of-flight aerosol chemical
speciation monitor (TOF-ACSM). The receptor site is geographically located in the middle part
of China, at the junction of the BTH region and the GZB region, which are the two of the three
key regions in Protection of Blue Sky issued by the National Congress for pollution control and
sustainable development in 2018. In addition, the chemical composition of non-refractory $PM_{2.5}$
(organics, sulphate, nitrate, ammonium, and chloride) and OA source apportionment were
resolved and analyzed with measured black carbon, gas-phase pollutants ($SO_2$, CO, $NO_2$ and
$O_3$) and meteorological parameters to provide complementary mass-based characterization of
the transported aerosols.
**2 Experimental**
**2.1 Sampling site and instrumentation**
The sampling was carried out on the rooftop of Le Méridien hotel, which was a 33-floor tall
building and about 200 meter above the ground (34.34°N, 109.02°E), during summer from 19th
May to 18th June 2018. It is located in the central area of Chan-ba Ecological District (CBE,
$km^2$), which was a new ecological district, located at the eastern part of the GZB region.
The sampling site was surrounded by wetlands and lawns.
A TOF-ACSM (Aerodyne Research Inc., Billerica, MA) was deployed in an air-conditioned
room on the top floor ($32^{nd}$) of Le Méridien hotel for continuous on-line measurements of non-
refractory $PM_{2.5}$ species including organics (Org), sulphate ($SO_4^{2-}$), nitrate ($NO_3^-$), ammonium
($NH_4^+$), and chloride ($Cl^-$). The sampling time resolution was 5 minutes. Also, a scanning
mobility particles sizer with a differential mobility analyzer (SMPS, model 3080) and a
condensation particle counter (CPC, model 3772) (TSI Incorporated, Shoreview, Minnesota,
USA) were combined for the particle number size distribution measurement between 10 ~ 840
nm, which shared an inlet with TOF-ACSM through a $PM_{2.5}$ cyclone (URG-2000-30ED, URG
Corp., Chapel Hill, NC). Black carbon concentration was measured by an aethalometer (AE33,
Magee Scientific) through an individual $PM_{2.5}$ cyclone (SCC, BGI) inlet. The sampling time-
resolution was 1 min at a flow rate of 5 L min$^{-1}$. Gas-phase pollutants ($SO_2$, CO, NO, $NO_2$ and
$O_3$) were measured by the gas analyzers (Thermo Scientific Inc.). Meteorological data
(temperature, RH, wind speed and wind direction) were measured by an automatic weather
station (MAWS201, Vaisala, Vantaa, Finland) and a wind sensor (Vaisala Model QMW101-
M2). All ambient inlets of instruments were set on the rooftop ($33^{rd}$, 200 m) and were 1.5 m in
height.
**2.2 TOF-ACSM operation**
TOF-ACSM has been detailed previously (Fröhlich et al., 2013). Briefly, ambient air was
sampled through a $PM_{2.5}$ cyclone and a 3/8-inch polished stainless-steel tube (Swagelok
company, Solon, OH) with a constant flow rate of 3 L min$^{-1}$ (0.3 L min$^{-1}$ for SMPS and CPC,
0.08 L min$^{-1}$ for TOF-ACSM and 2.62 L min$^{-1}$ for an extra constant flow air pump) for the
coarse particles cut. Following that, particles were focused into a narrow particle beam via a
$PM_{2.5}$ aerodynamic lens. Then the particles were evaporated by a thermal standard vaporizer (~



600°C) and ionized by an electron impact ionization (70eV), and the resulting ion fragments
were analyzed and determined by a time-of-flight mass spectrometer. Also, a Nafion dryer was
used to remove moisture prior to entering TOF-ACSM and SMPS, which kept the relative
humidity (RH) of the particle beam under 30%. Meanwhile, an automatically switching valve
was installed on the main air path between the Nafion dryer and TOF-ACSM, which was set to
change the sampling flow to a high-efficiency particulate air filter for the detection limits
measurement during the acquisition.
Ionization efficiency (IE) and relative ionization efficiency (RIE) calibrations were performed
about every ~10 days during the campaign. Briefly, pure ammonium nitrate and ammonium
sulphate particles were successively atomized by a TSI 3076 atomizer (TSI Incorporated,
Shoreview, Minnesota, USA). After that, they were dried by a hollow silica gel drying tube
before being imported into SMPS for 300 nm size selection, and then were counted and
measured by CPC and TOF-ACSM simultaneously. The other parameter calibrations, such as
the mass, the baseline, and the single ions were conducted every 3 days.

### 2.3 Data analysis

The chemical compositions and mass concentrations of $PM_{2.5}$ were analyzed by Tofware
(v2.5.13, Tofwerk AG). Organics, nitrate and chloride were analyzed with RIEs of 1.4, 1.1 and
1.3, respectively (Canagaratna et al., 2007). RIEs of ammonium and sulphate were estimated
from the averaged results of IE and RIE calibration (4.7 for RIE of ammonium; 0.67 for RIE of
sulphate). Besides, a particle collection efficiency (CE) for particle bounce losses was
calculated as a value of 0.5, with a slight adjustment of CE value was based on a composition
dependent collection efficiency (CDCE) approach following Middlebrook et al., 2012. The
resulting mass concentrations of chemicals of $PM_{2.5}$ were well correlated with the mass
concentrations of water-soluble inorganic aerosol from our In-situ Gas and Aerosol
Compositions monitor (IGAC, S-611, MachineShop) measurement (Fig. S2), suggesting the
reliability of TOF-ACSM results analysis.
The OA source apportionment was performed by positive matrix factorization (PMF, Paatero
and Tapper, 1994; Paatero, 1997) and multilinear engine (ME-2, Paatero, 1999). Organic
aerosol matrices (data matrix, error matrix, minimum values, time series and m/z from 1~120
amu in our case) were exported from Tofware, and were resolved for source apportionment in
PMF-ME-2 Toolkit SoFi (version 6.3, Canonaco et al, 2013). The optimal factor-selection and
constraining strategies of SoFi were described by Elser et al. (2016). The details are presented
in section S1 of the supplementary.

### 2.4 Trajectory analysis

The trajectory analysis was performed using the HYSPLIT model (Draxler and Hess, 1998) in
Hybrid Single-Particle Lagrangian Integrated Trajectory (HYSPLIT_4). Briefly, trajectories
were calculated every one hour from the air mass data which were downloaded from the
National     Oceanic     and     Atmospheric     Administration     (NOAA,
ftp://arlftp.arlhq.noaa.gov/pub/archives/gdas1) with 48 hours backward at a height of 200 m.
The trajectories were further clustered using in TrajStat (TrajStat_v1.2).



**2.5 Sulphur oxidation ratio and nitrate oxidation ratio**

Sulphur oxidation ratio (SOR) and nitrate oxidation ratio (NOR) are the ratios of sulphate and nitrate to their gaseous precursors, which were widely used to represent the degree of gas-to-particle conversions of sulphur and nitrogen. SOR and NOR are calculated by solving Eq. (1) and (2) (Ji et al., 2018; Chang et al., 2020).

$$SOR = n[SO_4^{2-}]/(n[SO_4^{2-}] + n[SO_2]) \tag{1}$$

$$NOR = n[NO_3^-]/(n[NO_3^-] + n[NO_2]) \tag{2}$$

**3 Results and discussion**

**3.1 Overview of the chemical composition, OA sources and regional transport in the receptor site**

The observational site with an altitude of ~200m above the ground provides ideal to investigate the impact of regional transport on aerosol properties. Figure 1 shows an overview of the time series of the chemical components of NR-PM$_{2.5}$ (Organic, sulphate, nitrate, ammonium and chloride), together with meteorological parameters and gas-phase pollutants (SO$_2$, CO, NO$_2$ and O$_3$).The average mass concentration of NR-PM$_{2.5}$ was 21.5±14.9 μg m$^{-3}$, similar to the previous AMS/ACSM results in the western China (24.5 μg m$^{-3}$, Xu et al., 2014) and the southeastern China during summer (14.5-32.9 μg m$^{-3}$, Huang et al., 2012; Lee et al., 2013; Huang et al., 2013) but was lower than that in the northern China (41-80 μg m$^{-3}$ , Hu et al., 2013; Duan et al., 2020). Organics constituted the largest fraction of NR-PM$_{2.5}$ (35% or 7.5 μg m$^{-3}$), followed by sulphate (25% or 5.3 μg m$^{-3}$), nitrate (17.0% or 3.7 μg m$^{-3}$), ammonium (14% or 3.0 μg m$^{-3}$), BC (8% or 1.7 μg m$^{-3}$), and chloride (1%, 0.2 μg m$^{-3}$).

Figure 2 shows the results of winds field map, cluster-averaged backward trajectory and winds rose analyses. Four transport sectors were identified, including the Beijing-Tianjin-Hebei region (BTH, the east cluster, red), the northern China (the north cluster, magenta), the rural Guanzhong Basin region (GZB, the south cluster, green) and the urban GZB region (the west cluster, blue).

The BTH transport was featured by the long-distance air mass trajectories advected over the North China Plain with an average wind speed of 1.9±1.8 m s$^{-1}$. The BTH transport sector accounted for 7% of the total observation days. It showed the highest mass concentration of PM$_{2.5}$ (32.9±17.4 μg m$^{-3}$).

The northern China transport sector was clustered by the transport from the Mongolia and the northern part of China, including Inner Mongolia and northern Shaanxi province. It represented the longest transport distance with an average wind speed of 2.2±2.1 m s$^{-1}$ and accounted for 22% of observation days. The PM$_{2.5}$ mass in the northern China transport sector was 24.9±12.9 μg m$^{-3}$, which was lower than that in the BTH transport sector.

The urban GZB transport sector was from the west of the GZB region, including those large cities in the GZB region, such as Baoji, Xianyang and Xi'an. The urban GZB transport sector was the most frequent pathway during the campaign, accounting for 60% of observation days



with an average wind speed of $1.0\pm0.9$ m s$^{-1}$. The PM$_{2.5}$ mass in the urban GZB transport sector
was $21.7\pm14.8$ µg m$^{-3}$. Finally, the rural GZB transport sector mainly consisted of the air mass
from Mt. Qinling, representing the air mass with least anthropogenic influence and accounting
for 11% of observation days with an average wind speed of $1.9\pm0.7$ m s$^{-1}$ and the lowest average
PM$_{2.5}$ mass ($8.8\pm5.5$ µg m$^{-3}$).

**3.2 Secondary inorganic formation during the transport**

Figure 3 shows the mass concentrations of the measured components, their fractional
contributions, the sulphur oxidation ratio (SOR) and the nitrogen oxidation ratio (NOR) in these
four transport sectors. SIA showed the highest mass concentration of $21.4\pm11.9$ µg m$^{-3}$ in the
BTH transport sector, followed by the northern China transport sector ($15.2\pm6.6$ µg m$^{-3}$), the
urban GZB transport sector ($12.2\pm3.1$ µg m$^{-3}$) and the rural GZB transport sector ($3.5\pm1.7$ µg
m$^{-3}$). The corresponding fractional contributions of SIA to PM$_{2.5}$ were 64%, 60%, 55%, and
39%. The difference in SIA mass and fractional contributions suggests the difference in SIA
precursor concentrations (i.e., SO$_2$, NO$_X$ and NH$_3$) and SIA formation efficiency among
different transport sectors, as discussed below.
Sulphate was the dominant fraction in the BTH transport sector, accounting for 30% of PM$_{2.5}$.
This fraction decreased to 25% and 24% in the northern China transport sector and the urban
GZB transport sector, respectively. Nitrate showed no obvious difference in the three urban
transport sectors, accounting for 17-19% of PM$_{2.5}$. For the rural GZB transport sector, the
fraction of sulphate and nitrate largely decreased to 19% and 11% of PM$_{2.5}$ respectively,
consistent with lower SO$_2$ ($3.2\pm2.5$ µg m$^{-3}$) and NO$_2$ ($27.8\pm10.3$ µg m$^{-3}$) in the rural GZB
transport sector which was about half of that in the three urban transport sectors (6.3-7.3 µg m$^{-3}$
for SO$_2$ and 44.7-51.3 µg m$^{-3}$ for NO$_2$). High fraction of sulphate in the BTH transport sector
was supported by high concentrations of SO$_2$ and sulphate in the BTH region and central China
region (Du et al., 2019; Chen et al., 2020). It was further supported by high sulphur conversion
efficiency (SOR), for which the BTH transport sector showed the highest SOR of 0.58, followed
by the northern China transport sector (0.52), the urban GZB transport sector (0.49), and the
rural GZB transport sector (0.44) (Figure 3c). Similarly, NOR showed relatively high valve of
0.29 in the BTH transport sector and the northern China transport sector, and was slightly low
in the urban GZB transport sector (0.25) and the rural GZB transport sector (0.24), consistent
with high nitrate fraction in the BTH transport sector and the northern China sector (Figure 3d).
In comparison with the previously reported results which were investigated in the source
regions of the urban GZB transport sector and the BTH transport sector (Xu et al., 2019; Duan
et al., 2020), SOR and NOR showed obvious increase after transport. For SOR it increased from
0.36 to 0.44 in the urban GZB transport pathway and from 0.53 to 0.58 in the BTH transport
pathway, while for NOR it increased from 0.06 to 0.25 in the urban GZB transport pathway and
from 0.15 to 0.29 in the BTH transport pathway. The increases in SOR and NOR after transport
suggest the efficient sulphate and nitrate formation during the regional transport. This was also
reflected in the sulphate and nitrate fractions (Figure 4). After transport the fractional
contribution of sulphate increased from 17% (3.8 µg m$^{-3}$) to 26% (5.6 µg m$^{-3}$) in the urban GZB
transport pathway and from 20% (6.2 µg m$^{-3}$) to 32% (9.9 µg m$^{-3}$) in the BTH transport pathway,
while the nitrate fraction increased from 12% (2.7 µg m$^{-3}$) to 19% (3.7 µg m$^{-3}$) in the urban



GZB transport pathway but slightly decreased from 24% (7.4 µg m$^{-3}$) to 19% (5.9 µg m$^{-3}$) in
the BTH transport pathway likely due to the volatilization of NH$_4$NO$_3$ during the long-distance
transport. We also compared the pollution episodes caused by the continuous transport from the
BTH (EP1) and the urban GZB (EP2) (as shown in the shaded area in Figure 1, detailed in Fig.
S3). Sulphate and nitrate were normalized by BC to minimize the influence of primary emission
or dilution (Figure 5). Sulphate/BC ratio increased with the transport in both EP1 and EP2, with
a growth rate of 0.26 hr$^{-1}$ during EP1 (increased from 1.2 to 9.4 in 31 hours) and of 0.1 hr$^{-1}$
during EP2 (increased from 2.3 to 16.2 in 131 hours). Nitrate/BC ratio showed a growth rate of
0.17 hr$^{-1}$ during EP1 (increased from 1.3 to 6.6 in 31 hours) and of 5.7 times lower during EP2
(0.03 hr$^{-1}$, increased from 1.1 to 4.7 in 131 hours). The comparison of these two episodes further
supports stronger formation of SIA in the BTH transport sector. The difference in the formation
efficiency of sulphate and nitrate in different transport air masses may be related to RH, because
aqueous-phase oxidation was an important formation pathway for sulphate at high RH
condition (Cheng et al., 2016; Xue et al., 2019; Chang et al., 2020) and high RH also
strengthened the conversion of gas-phase NH$_4$NO$_3$ to particle phase (Huang et al., 2020), which
likely leads to high SOR and NOR in the BTH transport sector (81±17% of average RH).

**3.3 Secondary organic formation during the transport**

Figure 6 shows the mass concentrations of the resolved OA factors, their fractional
contributions, the $f44$ versus $f43$ ratio and O/C ratio in these four transport sectors. $f44/f43$ ratio
and O/C ratio are important indicators of the oxidation state of bulk OA (Ng et al., 2010), which
were widely used in previous studies for SOA oxidation analysis (Xu et al., 2014; Canonaco et
al., 2015; Reyes-Villegas et al., 2016). The BTH transport sector showed the highest OA mass
concentration of 8.9±5.1 µg m$^{-3}$, followed by the urban GZB transport sector (7.3±4.0 µg m$^{-3}$),
the northern China transport sector (6.9±3.9 µg m$^{-3}$) and the rural GZB transport sector (4.6±
2.5 µg m$^{-3}$). The corresponding fractional contributions of MO-OOA to total OA were 58% (5.2
µg m$^{-3}$), 55% (4.0 µg m$^{-3}$), 57% (4.0 µg m$^{-3}$), and 42% (1.9 µg m$^{-3}$), constituting the major OA
source in the four transport sectors. The LO-OOA fraction was higher in the rural GZB transport
sector (34%, 1.9 µg m$^{-3}$) compared to the other three urban transport sectors (around 23%, 3.9-
5.2 µg m$^{-3}$) suggesting that SOA was less oxidized in the rural transport sector likely due to
large emission of biogenic VOCs from the Mt. Qinling area. The northern China transport sector
showed the highest $f44/f43$ ratio of 2.1 and O/C ratio of 0.87, followed by the BTH transport
sector (1.9 and 0.78), the urban GZB transport sector (1.8 and 0.72), and much lower values in
the rural GZB transport sector (1.6 and 0.58). The higher $f44/f43$ and O/C ratio in the northern
China transport sector and the BTH transport sector suggests sufficient OA aging during long-
range transport. The $f44/f43$ ratios in these four transport sectors were higher than those in the
urban sites in previous studies (triangle in Figure 6c, Ng et al., 2011) and the O/C ratios in these
four transport sectors (0.72-0.87) were also much higher than those measured in urban sites in
China during summer, such as Lanzhou (0.33, Xu et al., 2014) and Jiaxing (0.28, Huang et al.,
2013), but similar to the result from the long-range transport study in the United States (0.8,
Gunsch et al., 2018). Note that the O/C ratios in the transport sectors were also much higher
than those measured in the source regions of the urban GZB transport sector and the BTH
transport sector, with O/C ratio increasing from 0.54 to 0.78 in the BTH transport pathway and





from 0.58 to 0.72 in the urban GZB transport pathway after transport. The corresponding MO-OOA to SOA fraction also increased from 37% (3.4 μg m⁻³) to 72% (5.2 μg m⁻³) in the BTH transport pathway and from 37% (3.6 μg m⁻³) to 70% (4.0 μg m⁻³) in the urban GZB transport pathway (Figure 7), suggesting regional transport enhanced OA aging process and thus the OA oxidation state. The growth rates of MO-OOA and LO-OOA during the pollution episodes of EP1 and EP2 are shown in Figure 8. Similar to SIA, MO-OOA/BC ratio increased with the transport duration for both episodes. It showed a growth rate of 0.15 hr⁻¹ during EP1 (increased from 0.23 to 4.77 in 31 hours) and of 0.06 hr⁻¹ during EP2 (increased from 1.58 to 9.59 in 131 hours), suggesting stronger formation of MO-OOA in the BTH transport sector. On the contrary, LO-OOA showed no obvious increasing trend with the transport duration during EP1 and EP2, likely due to a higher conversion efficiency from LO-OOA to MO-OOA.

**4. Conclusion**

The observation at ~200 m above the ground in the junction of North China Plain and Fenwei Basin showed that the fraction of SIA and MO-OOA increased significantly after transport. The sulfur oxidation rate (SOR, 0.49-0.58), nitrogen oxidation rate (NOR, 0.25-0.29), $f44/f43$ ratio (1.6-2.1) and O/C ratio (0.72-0.87) were significantly higher than those investigated locally, indicating that long-distance transport largely enhanced the SIA formation, the OA oxidation and aging. The formation rate of sulphate, nitrate and MO-OOA in the BTH transport sector was much higher than that in the GZB transport sector, indicating the stronger sulphate, nitrate and MO-OOA formation efficiency in the BTH transport sector.

**5. Data availability**

The detailed data can be obtained from https://doi.org/10.5281/zenodo.6446514 (Zhong et al., 2022).

**Acknowledgement**

This work was supported by the National Natural Science Foundation of China (NSFC) under Grant No. 41925015 and 41877408, the Chinese Academy of Sciences (no. ZDBS-LY-DQC001), and the Cross Innovative Team fund from the State Key Laboratory of Loess and Quaternary Geology (No. SKLLQGTD1801).

**Author contributions**

**Haobin Zhong**: Methodology, data curation, Formal analysis, Writing – original draft, Writing – review & editing. **Ru-jin Huang**: Conceptualization, Validation, Data curation, Writing – original draft, Writing – review & editing, Supervision, Project administration, Funding acquisition. **Chunshui Lin**: Writing - review & editing. **Wei Xu**: Writing - review & editing. **Jing Duan**: Writing - review & editing. **Yifang Gu**: Writing - review & editing. **Wei Huang**: Writing - review & editing. **Haiyan Ni**: Writing - review & editing. **Chongshu Zhu**: Resources. **Yan You**: Writing - review & editing. **Yunfei Wu**: Resources. **Renjian Zhang**: Resources. **Jurgita Ovadnevaite**: Writing - review & editing. **Darius Ceburnis**: Writing - review &





editing. **Colin D. O'Dowd**: Writing - review & editing.
**Competing interests**
The authors have no competing interests to declare.

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

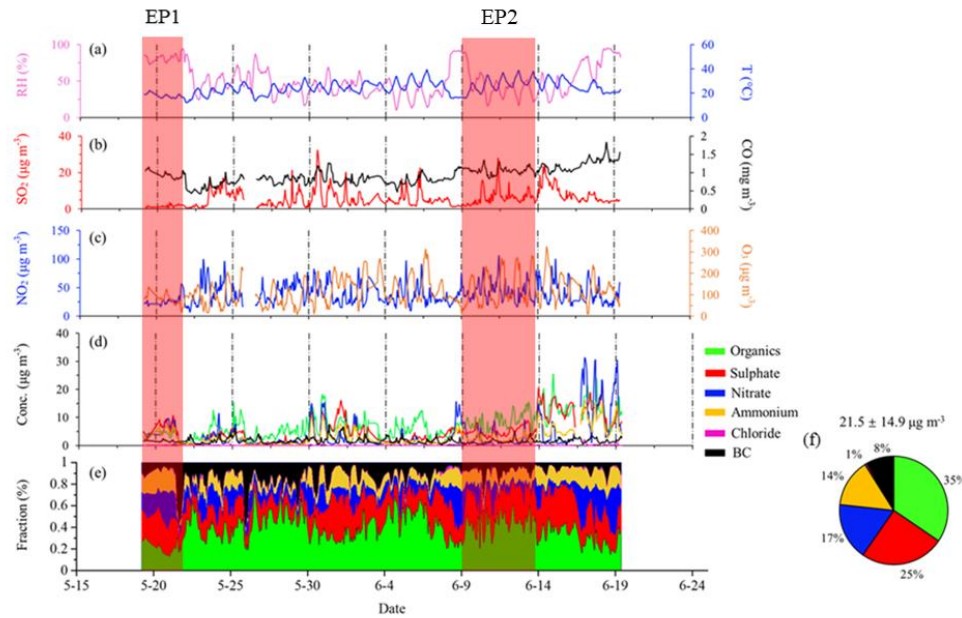


**Figure 1.** Time series of (a) relative humidity and temperature, (b, c) mass concentration of
SO$_2$, CO, NO$_2$ and O$_3$ (d, e) mass concentrations and fractional contributions of PM$_{2.5}$ (organics,
sulphate, nitrate, ammonium, chloride and BC) during the campaign period. Five pollution
episodes are observed during the entire campaign, and they are detailly analyzed by HYSPLIT
model and showed in Fig. S3. EP1 and EP2 (shaded) are the only two pollution episodes caused
by continuous transport from the BTH transport and the urban GZB transport, respectively.
Therefore, they are selected for further discussion.



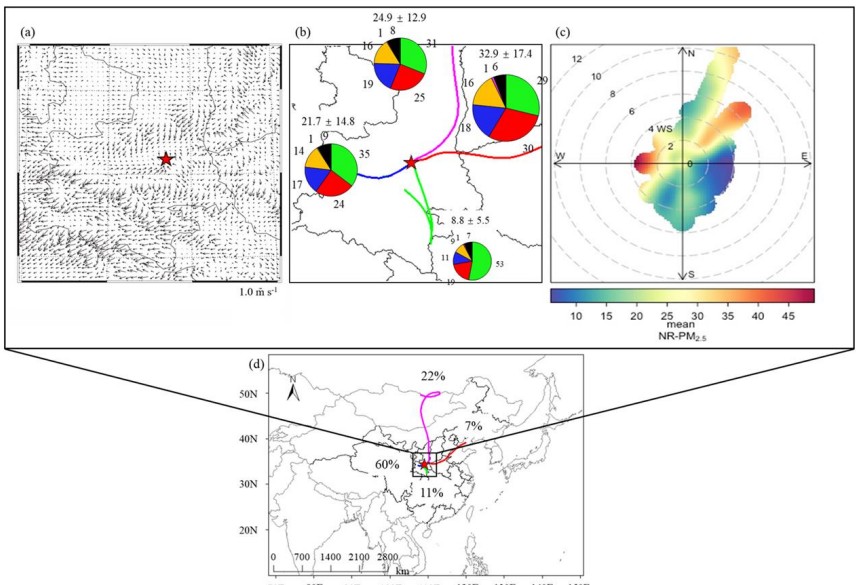

**Figure 2.** (a) Wind field, (b, d) backward trajectory and (c) wind rose results during the campaign. There are four transport clusters observed during the campaign, which are the northern China transport (the north cluster, magenta, 22% of observing days) and the BTH transport (the east cluster, red, 7% of observing days), the western GZB transport (the west cluster, blue, 60% of observing days) and the southern GZB transport (the south cluster, green, 11% of observing days).



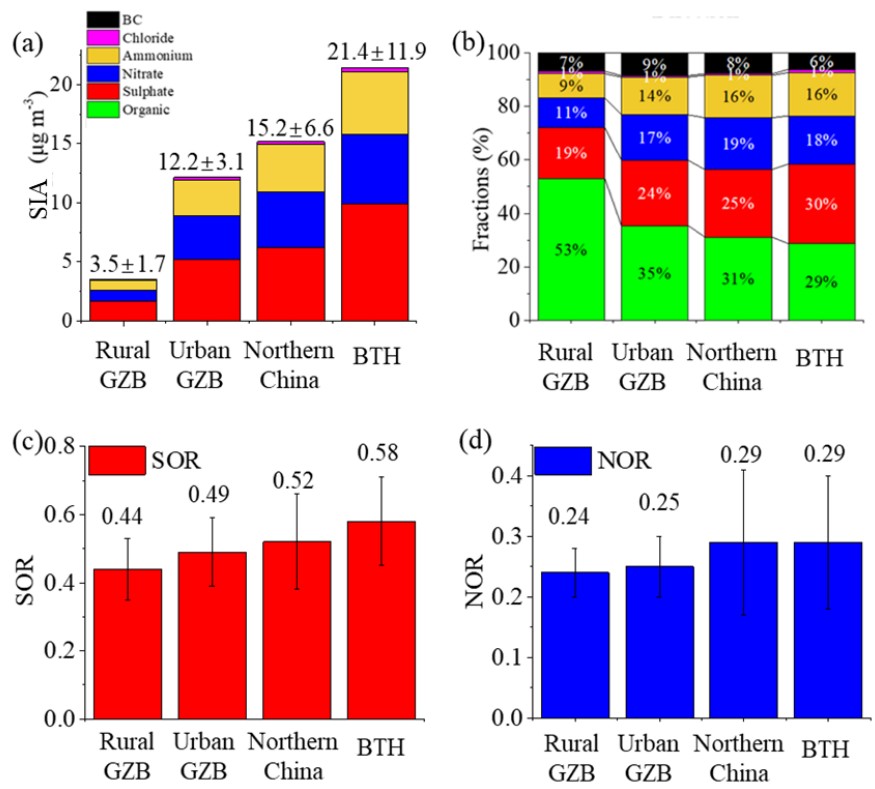

**Figure 3.** The comparison of (a) the mass concentration of SIA, (b) chemical fractions of PM$_{2.5}$, (c) sulphur oxidation ratio (SOR) and (d) nitrogen oxidation ratio (NOR) in four transport sectors.

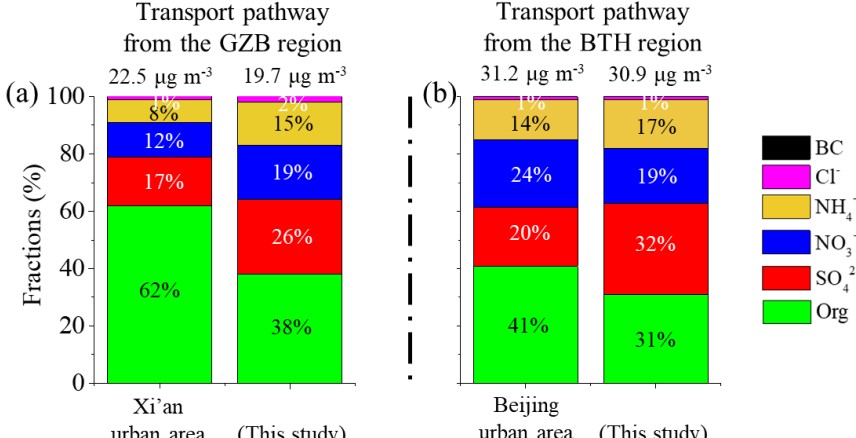

**Figure 4.** Chemical composition of the observing results which were long-term observation and were right on the transport route of the BTH transport and the GZB transport, including the Beijing urban area (Xu et al., 2019), the Xi'an urban area (Duan et al., 2020), the BTH transport in this study (East transport) and the urban GZB transport in this study (West transport).

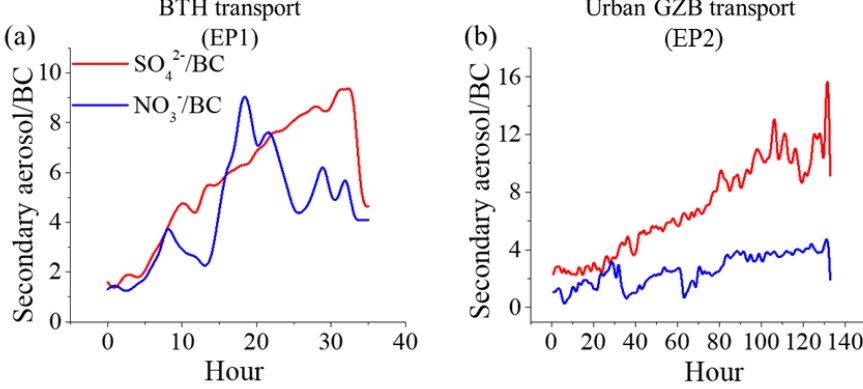

**Figure 5.** The relationship between production of the secondary inorganic aerosol and transport duration in the pollution episodes. EP1 and EP2 represented the pollution episodes caused by the BTH transport and the urban GZB transport, respectively.



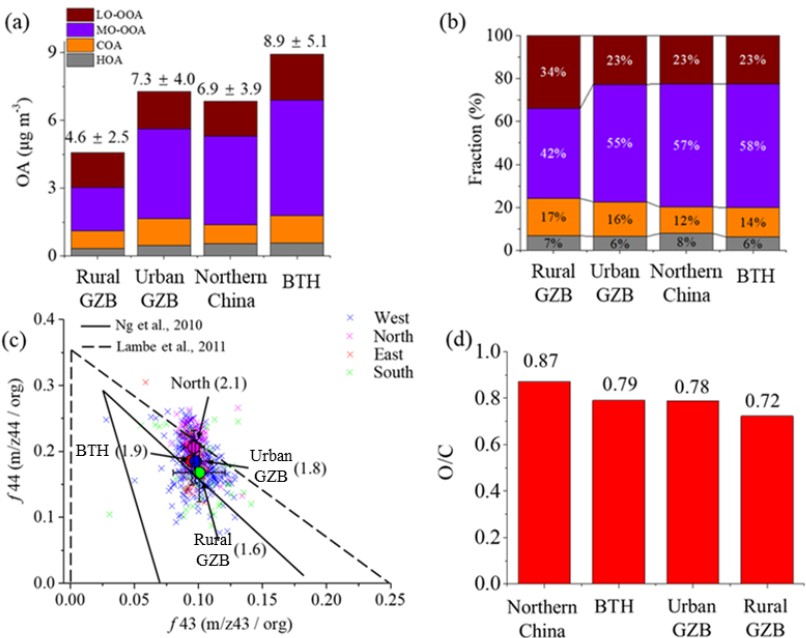

**Figure 6.** The comparison of (a) the mass concentration and (b) fractions of organic aerosol. (c)
Scatter plot of $f44$ v.s. $f43$ in four transport directions. The triangle from Ng et al., (2010) and
Lambe et al., (2011) is drawn in solid line and dotted line, respectively. (d) The O/C ratio in four
transport directions.

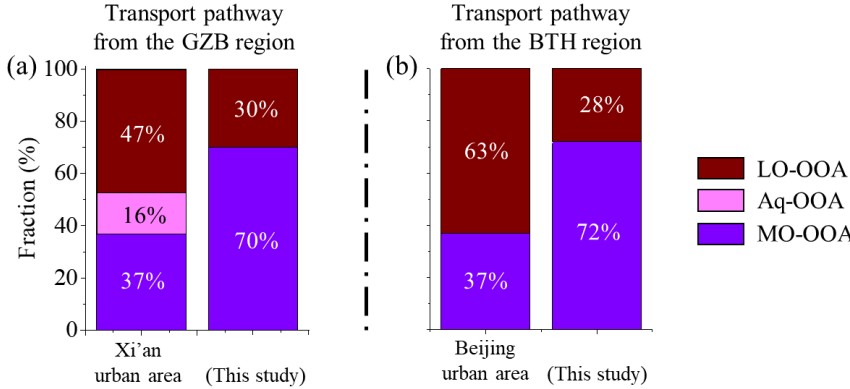

**Figure 7.** OA factors of the observing results which were long-term observation and were right
on the transport route of the BTH transport and the GZB transport, including the Beijing urban
area (Xu et al., 2019), the Xi'an urban area (Duan et al., 2020), the BTH transport in this study

(East transport) and the urban GZB transport in this study (West transport).

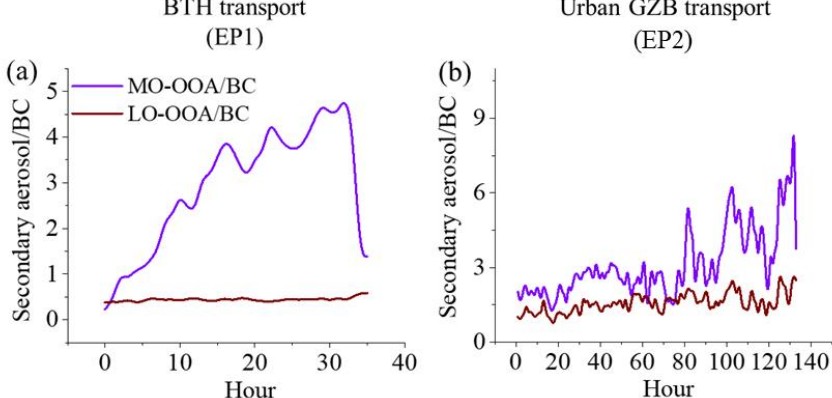

630

**Figure 8.** The relationship between production of the secondary organic aerosol and transport
duration in the pollution episodes. EP1 and EP2 represented the pollution episodes caused by
the BTH transport and the urban GZB transport, respectively.