# Peer review of "Measurement report: On the contribution of long-distance transport to the secondary aerosol formation and aging"

_Atmospheric Chemistry and Physics, 2022_

## Author Comment (AC1)

The authors thank the editor and referees to review our manuscript and particularly for the valuable comments and suggestions that are very helpful in improving the manuscript. We provide below point-by-point responses to the referees' comments. We also have made most of the changes suggested by the referees in the revised manuscript.

Reply on RC1

Referee #2

The paper titled "Measurement report: On the contribution of long-distance transport to the secondary aerosol formation and aging" by Zhong et al. present detail analysis in physio-chemical properties of aerosol transported from major pollution regions in North China at a regional receptor site located at the junction of the North China Plain and Fenwei Basin. The chemical composition of non-refractory $PM_{2.5}$ were measured by a Tof-ACSM and organic aerosol source apportionment were resolved and analyzed with measured black carbon, gas-phase pollutants and meteorological parameters to explore the secondary inorganic/organic formation during the transport. This study provides useful information on understanding the influence of long-distance transport of aerosol on the atmospheric environment. The manuscript was well written and presented, but some issues needed to be clarified. Therefore, I recommend the publication of Zhong et al. work after replying the following comments clearly.

**Response:** We thank the referee for the positive feedback on our manuscript, below we provide point-by-point reply to the comments.

1. Line 26-27, three or four pollution transport sectors? It seems four sectors as it descibe that Beijing-Tianjin- Hebei (BTH), urban Guanzhong Basin (GZB), northern China and one clean transport sector from rural Guanzhong Basin region were identified. Please confirm it and revised it properly.

**Response:** We thank for the referee's question. There are four transport sectors in our results, including three pollution transport sectors and one clean transport sector.

2. Line 45-47. Some recent studies about the air pollution in China are missed here, such as "Atmos. Chem. Phys., https://doi.org/10.5194/acp-18-8849-2018".

**Response:** We thank for the referee's advice. We have added the suggested article in Line 45-47.

3. Line 157, the height of 200 m is above the ground? As the observation site is on the rooftop of a 33-floor tall building which is already 200m above the ground, the height of back trajectories would be better than 200m. Please clarify it.

**Response:** We thank for the referee's question. In fact, we have tried to use different altitudes (including 200m, 300m, and 500m) during the trajectory calculation in HYSPLIT_4. By comparison, the clustering results in the Guanzhong Basin were more consistent with the results of wind field and wind rose chart when the altitude was set as 200m. Moreover, the 200m height

is also the actual observation height in our study, which provided more reasonable results of air mass transport at this altitude.

4. Line189-199. It is interesting to note that the average $PM_{2.5}$ concentration is higher in the northern China transport sector than that in the urban GZB transport sector, the latter sector usually has more anthropogenic emissions and more polluted than the former. Maybe the much higher frequency of pathway in urban GZB transport sector (60%) resulted in the lower $PM_{2.5}$ value. Please clarify it.

**Response:** We thank for the referee's question. As the referee mentioned that the urban GZB transport showed greater impact on the receptor site (60%), including many low $PM_{2.5}$ concentration events and resulting in lower mass concentration in average. However, the northern China transport showed relatively less impact on the receptor site (22%). Most of the air masses observed in the northern China transport were during the high wind speed period with higher pollutants contained, resulting in higher mass concentration in average. This might be the reason that the northern China transport showed higher mass concentration than the GZB transport.

5. Line 235-240. It is a good idea to compare the chemical composition between the receptor site and the transport sector. However, it should keep in mind that the difference in the chemical composition would be also originated by the different observation period, not only due to the transformation during the long-range transport. I note that the observation period in Beijing urban area by Xu et al., 2019 was almost the same as the present study, but the observation period in Xi'an urban area by Duan et al. 2021 (not Duan et al., 2020 in Figure 4) was different. In addition, as the BTH transport sector and GZB transport sector only contributed a part of the full observation period, it is better to select the same period to do the comparison. Thus, more information should be provided here.

**Response:** We thank for the referee's question. The comparison between the transport results and the reported results in the previous studies was simply used to prove the long-distance transport enhanced the SIA formation and SOA aging. However, there was no comparable result in the Guanzhong Basin during summer 2018, thus we used the result from Duan et al., 2021 which was the closest result (summer 2019) to our observation. Therefore, we were expecting there was no much difference between summer 2018 and summer 2019, which can reflect our conclusion to some extent. In addition, the comparison of the offline water-soluble sulphate (3.8 $\mu g\ m^{-3}$ in the urban GZB transport v.s. 4.6 $\mu g\ m^{-3}$ in receptor site), nitrate (2.5 $\mu g\ m^{-3}$ in the urban GZB transport v.s. 4.2 $\mu g\ m^{-3}$ in receptor site) and OC/EC ratio (3.3 in the urban GZB transport v.s. 4.1 in receptor site) sampled during the same period (summer in 2018) between the source region and the receptor site can also provide some support. We also thank the referee for pointing out the typo of Duan et al., 2020 in the manuscript and caption of Figure 4 and Figure 7. Now they were changed to Duan et al., 2021 in the revised manuscript.

6. Line 267-268 I do not understand the explain for the less oxidized SOA observed in the rural transport sector. The atmospheric oxidation of anthropogenic VOCs is prevailed over the biogenic VOCs? Please clarify it.

**Response:** We thank for the referee's question. We have no direct evidence to prove that the increased LO-OOA fraction in the rural GZB transport is related to the biogenic VOCs in the Mt.Qinling area. For better explanation, now it reads, "The LO-OOA fraction was higher in the rural GZB transport sector (34%, 1.9 μg m$^{-3}$) compared to the other three urban transport sectors (around 23%, 3.9-5.2 μg m$^{-3}$) suggesting that SOA was less oxidized in the rural transport sector."

7. Line 274-278. To back up the much higher O/C ratios observed in the transport sector, the O/C ratio results observed in the background area in North China would be suitable to explain the long-range transportation from source region to the receptor site. Such as the results reported in Li et al., 2021 (Atmos. Chem. Phys., https://doi.org/10.5194/acp-21-4521-2021) conducted at a mountainous site in North China Plain in summer 2019.

**Response:** We thank for the referee's suggestion. In line 277, now it reads, ", but similar to the results from the mountainous site in North China Plain during summer (0.75, Li et al., 2021) and the long-range transport study in the United States (0.8, Gunsch et al., 2018)." in the revised manuscript.

8. Line 278-281. The comparisons of O/C and SOA fractions between the receptor site (this study) and the source regions (previous studies) should be also noticed that the difference would be caused by the different observation period. More discussion should be provided before analysis the comparison results.

**Response:** We agree with the referee's opinion. However, there was no comparable result in the Guanzhong Basin during summer 2018, thus we used the result from Duan et al., 2021 which was the closest result (summer 2019) to our observation. We were expecting there was no much difference between summer 2018 and summer 2019, which can reflect our conclusion to some extent. In addition, the comparison of the offline OC/EC ratio (3.3 in the urban GZB transport v.s. 4.1 in receptor site) sampled during the same period (summer in 2018) between the source region and the receptor site can also provide some support.

Reply on RC2

Referee #1

This study performed in-situ measurements at a regional receptor site with the height of ~200 m above the ground, which locates in the junction of the North China Plain and Fenwei Basin, to investigate the influences of region-to-region transport on aerosol chemical compositions and secondary formations. The authors compared the characteristics of secondary aerosol species among the four transport sectors, finding that long-distance transport largely enhanced the SIA formation, the OA oxidation and aging. The manuscript is generally well written. I recommend for its publication after addressing the following comments.

**Response:** We thank the referee for the positive feedback on our manuscript, below we provide point-by-point reply to the comments.

1. Line 31-32, which aspect the MO-OOA played a dominant role in?

**Response:** We thank for the referee's question. MO-OOA played a dominant role in the source of organic aerosol. Now it reads, "…played a dominant role in the source of organic aerosol in all sectors including the clean one…"

2. Line 78-79, "we performed a two-months observation", but only one month data points were shown in Figure 1. Please confirm that.

**Response:** We thank for the referee's question. The actual observation period should be consistent with the data points in Figure 1. Now it reads, "we performed a one-month observation…" in the revised manuscript.

3. Line 100, SMPS is the abbreviation of scanning mobility particles sizer, and differential mobility analyzer should be abbreviated as DMA. In addition, the results of SMPS measurements are not given in the manuscript, please clarify it.

**Response:** We thank for the referee's advice. Now it reads, "…a differential mobility analyzer (DMA, model 3080) …" in the revised manuscript. SMPS was only used for calibration of ACSM in this study, thus there was no measurement result shown in the manuscript.

4. Line147, "m/z from 1~120" should be from12~120ï¼

**Response:** We thank for the referee's advice. Now it reads, "…m/z from 12~120 amus…" in the revised manuscript.

5. Line 224, valve?

**Response:** We thank for pointing out the typo. Now it reads, "…showed relatively high value of…" in the revised manuscript.

6. Line 219-221, the authors wanted to illustrate the high fraction of sulphate in the BTH transport sector was caused by high concentrations of $SO_2$ and sulphate in the BTH region, however, the citing reference here can not sufficiently support this conclusion. The more information should be given to prove the higher concentration level of $SO_2$ and sulphate in the BTH region than other sectors.

**Response:** We thank for the referee's question. We cited two additional studies to support this conclusion. Li et al (2021) showed high fraction of sulphate (37% of NR-$PM_1$) during summer in the NCP region. Sun et al (2022) suggested that high concentrations of aerosols were dominated by sulphate in the free troposphere during summer in the NCP region. The citing reference here showed higher $SO_2$ concentration and sulphate fraction than those reported in the source region of other sectors, for example sulphate fraction and $SO_2$ concentration was 37% of NR-$PM_1$ and 5.4 μg m$^{-3}$ in the NCP region, respectively (Li et al., 2021), higher than those of 17% of NR-$PM_{2.5}$ and 4.6 μg m$^{-3}$ in the GZB region (Duan et al., 2021). We hope these two references here can provide additional support.

Li, J., Cao, L., Gao, W., He, L., Yan, Y., He, Y., Pan, Y., Ji, D., Liu, Z., and Wang, Y.: Seasonal variations in the highly time-resolved aerosol composition, sources and chemical processes

of background submicron particles in the North China Plain, Atmos. Chem. Phys., 21, 4521–4539, https://doi.org/10.5194/acp-21-4521-2021, 2021.

Sun, P., Nie, W., Chi, X., Huang, X., Ren, C., Xue, L., Shan, Y., Wen L., Li, H., Chen, T., Qi, Y., Gao, J., Zhang, Q., and Ding, A.: Aircraft study of secondary aerosols in long-range transported air masses from the North China Plain by a mid-latitude cyclone. Journal of Geophysical Research: Atmospheres, 127, e2021JD036178, https://doi.org/10.1029/2021JD0361, 2022.

7. Line 239-241, some recent studies on evolutions of aerosol physicochemical properties during the transport processes, suggesting the mass loss of semi-volatile aerosol species driven by the evaporation process when aerosols are exposed to a cleaner environment, such as "Atmos. Chem. Phys., https://doi.org/10.5194/acp-21-14749-2021". It would be suitable to explain the decreased nitrate faction in the BTH transport pathway.

**Response:** We thank for the referee's advice. Now it reads, "…likely due to the mass loss of semi-volatile aerosol species (such as $NH_4NO_3$) when aerosols are exposed to a cleaner environment during the long-distance transport (Liu et al., 2021)."

8. Line 265-268. Is the statement here means the LO-OOA at this site were mainly contributed by biogenic VOCs oxidation? Please provide more information to clarify it.

**Response:** We thank for the referee's question. We have no direct evidence to prove that the increased LO-OOA fraction in the rural GZB transport is related to the biogenic VOCs in the Mt.Qinling area. For better explanation, now it reads, "The LO-OOA fraction was higher in the rural GZB transport sector (34%, 1.9 μg m$^{-3}$) compared to the other three urban transport sectors (around 23%, 3.9-5.2 μg m$^{-3}$) suggesting that SOA was less oxidized in the rural transport sector."